# The Influence of a Girls’ Health and Well-Being Program on Body Image, Self-Esteem, and Physical Activity Enjoyment

**DOI:** 10.3390/bs13090783

**Published:** 2023-09-21

**Authors:** Kellie Walters, Chrissy Chard, Esmeralda Castro, Devin Nelson

**Affiliations:** 1Department of Kinesiology, California State University, Long Beach, CA 90840, USA; 2Department of Health and Exercise Science, Colorado State University, Fort Collins, CO 80523, USA; 3Department of Counseling Psychology & Human Services, University of Oregon, Eugene, OR 97403, USA; ecastro@uoregon.edu

**Keywords:** adolescence, female, psychosocial health, resistance training, physical activity

## Abstract

Adolescent girls report low physical activity levels and poor body image and self-esteem. This study evaluated the impact of a girls’ health intervention on body image, self-esteem, and physical activity enjoyment. The intervention was grounded in self-determination theory, resulting in lessons designed to promote autonomy, competence, and relatedness. The two primary components of the intervention included opportunities for girls to learn about resistance training as well as ways to improve their psychosocial health (i.e., body image, self-esteem, and physical activity enjoyment). Girls (*n* = 590), in the intervention (*M_age_* = 12.79, *SD* = 0.69) and control group (*M_age_* = 12.92, *SD* = 0.73), completed pre and post measures. A repeated measures MANOVA was conducted to assess changes in body image, self-esteem, and physical activity enjoyment. The intervention resulted in a significant increase in body esteem-appearance, *F* = 9.23, *p* = 0.003, and body esteem-weight, *F* = 4.77, *p* = 0.029, and a greater, non-significant, increase in self-esteem (3.22%), and physical activity enjoyment (4.01%) compared to the control group. This highlighted the use of the intervention for significant improvements in appearance and weight-related body image. The results support implementing psychosocial lessons, as well as physical activity, in health programming for girls.

## 1. Introduction

The benefits of physical activity are highly recognized and promoted throughout the public health community. The Physical Activity Guidelines for Americans, 2nd edition provides evidence-based physical activity recommendations and outlines the supported short- and long-term benefits that can occur from participating in physical activity. In youth, the benefits can include, but are not limited to, improved cognition and physical health and reduced risk of depression [1,2]. According to the guidelines, youth are recommended to participate in at least 60 min of moderate-to-vigorous intensity, mostly aerobic, physical activity (MVPA) each day. Also included are recommendations for participating in at least three days of vigorous-intensity physical activity and muscle- and bone-strengthening activities.

Despite the widely known benefits of physical activity, adolescent girls are unlikely to meet the recommended amount of physical activity [1]. This disparity has been shown to persist into adulthood [3], increasing the risk for chronic disease. Cooper et al. [4] found that physical activity rates declined 4.6% each year among adolescent girls, while Farooq et al. [5] reported physical activity levels declined 5.3% a year among girls aged three through 18. These results highlight the need for tailored programs aimed to encourage physical activity among this population. Due to this evident disparity, it has been suggested that additional support and encouragement among adolescent girls may be needed for physical activity [1].

During adolescence, girls commonly experience detriments to their psychosocial health, a multidimensional construct including one’s physical, social, mental, and emotional health [6]. This includes poor body image, psychological distress (e.g., depression and anxiety), and poor self-esteem [7]. Adolescence is also a critical point of transition in which girls no longer find participation in sports and physical activity enjoyable or a priority [8]. Common reasons for the observed decline in physical activity among adolescent girls include negative experiences in school, such as peer teasing [9] and low enthusiasm from physical education teachers regarding girls’ sports [10]. Girls also report self-perceived lack in capability for physical activities [10] and concerns in body image [9]. As regular participation in physical activity among adolescent girls reduces the risk of developing chronic disease, is positively associated with academic performance, and reduces the risk for depression [1], it is important to promote positive physical activity behavior among adolescent girls.

### 1.1. Self-Determination Theory

Self-determination theory (SDT) is a theory of motivation commonly used in developing programs or interventions geared toward youth [11]. SDT posits that humans have an innate need for psychological growth and in order to achieve optimal psychological well-being and facilitate intrinsic motivation [12], humans need to experience autonomy, competence, and relatedness [13]. Autonomy describes the need to act freely rather than acting against own’s own free will, competence is defined as the need to demonstrate effectiveness or impact in a given situation, and relatedness is the need to feel a sense of connection and belonging [14]. A recent meta-analysis reported that interventions informed by SDT resulted in improved health behaviors (including physical activity) at the end of the intervention and in follow-up measures [15]. If youth are given more opportunities to experience autonomy, competence, and relatedness, their level of self-determined motivation increases which is particularly important for long-term behavior adoption (e.g., engaging in physical activity as they age).

### 1.2. Body Image

Body image is defined as an individual’s own perception of their body, influenced by various factors (e.g., cultural norms, attitudes of weight and shape) [16]. Establishing a positive perception of one’s body image can be challenging for adolescent girls [17,18]. Research shows that among adolescent girls, the transition from youth to adolescence is a pivotal time during which girls develop negative attitudes toward body image [19]. These attitudes are often attributed to physical changes that result from puberty, as well as the increased importance of peer acceptance and social comparisons [20].

Adolescent girls are more likely to experience poorer body perceptions compared to their male counterparts, most likely as a result of cultural and social emphasis regarding physical appearance [21]. The widespread acceptance that has been directed toward a thin body shape among women can make it increasingly difficult for girls to accept their bodies as they are, especially during puberty-related changes [22]. Body dissatisfaction poses a major health threat to adolescent girls as it is associated with impaired emotional well-being, low self-esteem, anxiety and depressive symptoms, low levels of physical activity, and increased risk for disordered eating [23,24]. Andrew et al. [25] found that body dissatisfaction served as a risk for lower self-esteem among girls, particularly during early adolescence. Early interventions that promote positive body image among adolescent girls may aid in reducing the rate of psychological distress and body dissatisfaction [17].

Adverse outcomes associated with negative body image among adolescent girls highlight the need for early intervention to promote positive body image and healthy behaviors, such as physical activity. One approach to “protecting” adolescent girls’ body image is through physical activity. Adolescents who participate in regular physical activity report higher levels of body satisfaction [7]. Research also demonstrates that having a positive body image is a strong, statistical predictor of greater physical activity participation in adolescent girls [26]. Therefore, reasons to encourage adolescent girls to participate in physical activity go beyond the benefits related exclusively to physical health.

### 1.3. Self-Esteem

Self-esteem is one’s own sense of self-worth and confidence in one’s abilities. Self-esteem has shown to decline in girls during adolescence, and girls also experience lower self-esteem than their male peers [2,21]. Research suggests that bodily changes that occur during puberty play a significant role in reducing the number of positive thoughts girls have about themselves during adolescence [27]. Low self-esteem is associated with poorer physical and mental health [24], and reduced physical activity engagement [28]. However, engaging in physical activity can also improve self-esteem [7], which is important because, recently, data suggests that self-esteem acts as a mechanism of resilience to adolescents exposed to stress [29]. In a cross-sectional study analyzing the interaction between stress, self-esteem, and physical activity in 6504 adolescents, Carter (2018) reported that those adolescents with high self-esteem were more likely to deal with daily stressors successfully and maintain their physical activity behaviors over time.

### 1.4. Physical Activity Enjoyment

Physical activity enjoyment can serve as a significant determinant of physical activity behaviors among adolescent girls and is characterized by fun and pleasure related to engaging in physical activity [30]. Physical activity enjoyment has been associated with adolescent girls’ level of self-consciousness about their body, where there is a negative association in girls with lower psychological well-being [31]. This, in turn, is related to participating in lower amounts of moderate-to-vigorous physical activity [31]. Additionally, adolescent girls who enjoy physical activity are often more intrinsically motivated to perform physical activity, which also serves as an important facilitator for sustained physical activity participation [32]. Intrinsic motivation, such as finding pleasure in the activity, has been shown to be more important for sustaining behaviors compared to motivation that is extrinsic, which relies heavily on external rewards [33]. The more that girls enjoy physical activity, the more likely they are to participate in physical activities. Importantly, physical activity enjoyment has been shown to attenuate the decline in physical activity seen as youth become older [34] and serves as a facilitator for continued engagement of physical activity behaviors into adulthood [33].

Adolescent girls remain one of the most health-disparate populations [1,7]. Based on the relationship between physical activity participation and psychosocial health among adolescent girls, greater efforts should be put towards creating theory-informed health promotion programs that encourage physical activity and promote body image, self-esteem, and physical activity enjoyment among adolescent girls [35,36]. This research is innovative because it is the first study to assess an intervention aimed at improving adolescent girls’ health through two mechanisms: physical activity (i.e., resistance training) and psychosocial health. Therefore, the purpose of this study is to quantitatively evaluate the efficacy of a novel health intervention for adolescent girls on increasing positive body image, self-esteem, and physical activity enjoyment. We hypothesized that the intervention will significantly improve participants’ body image, self-esteem, and physical activity enjoyment.

## 2. Material and Methods

### 2.1. Sample

From spring 2015 to spring 2016, data were collected at a middle school in South Carolina where the intervention was being offered. There were two different cohorts: (1) an intervention group which was comprised of girls who completed at least 75% or more of the intervention and (2) a control group, which was comprised of girls who did not participate in intervention. The intervention group was recruited from a list of middle school girls attending a middle school in South Carolina who were already signed up for a community intervention (described in detail below). The control group was recruited from the same middle school as the experimental group and included girls who were not participating in the intervention. The intervention group included 58 adolescent girls (*M_age_* = 12.79, *SD* = 0.69) and had a 100% retention rate. The control group included 532 adolescent girls (*M_age_* = 12.92, *SD* = 0.73). The sample sizes were different between groups because of the difference in inclusion requirements, which has been noted as a methodological issue in community-based interventions [37]. Logistically, the number of girls who could participate in the intervention was limited due to the coach/participant ratio (1:7) and access to facility space and equipment. The control group was comprised of a much larger sample size because it was sampled from the remainder of girls who did not participate in the intervention but attended the same school as those girls in the intervention. The questionnaire took approximately 15–20 min to complete and was completed twice with 10 weeks between each survey completion. For the intervention group, data collection occurred at the start of the program and ten weeks later at the conclusion of the program. Data for the control group were collected 10 weeks apart during the participants’ physical education classes. Parent consent and child assent were received prior to collecting data and all methods were approved by the first author’s Institutional Review Board and the associated school districts.

### 2.2. Intervention

During the intervention, adolescent girls participated in a licensed curriculum: Smart Fit Girls (SFG). To protect the intellectual property of SFG, limited details about the program are available to the public, including academic journals. However, a logic model outlining the structure of the program can be found in Walters, Chard, Jordan, and Anderson [17], where they qualitatively examined the impact of the intervention on body image. A brief description of the program is detailed below and an infographic representing the various components of the intervention can be found in Figure 1.

SFG is a 10-week after-school empowerment program that promotes physical, mental, and emotional well-being in middle-school-aged girls [17]. During the program, girls participated in a variety of activities aimed to improve their physical activity enjoyment, body esteem, and self-esteem. During each SFG session, participants engaged in 20–30 min of physical activity and 60–90 min of lessons geared towards improving psychosocial health. The time participants spent being physically active was in alignment with current physical activity recommendations for youth [1]. The primary modality of physical activity the girls participated in was strength training which was purposefully chosen to empower the girls to embrace their strengths and what their bodies can do when they are strong (body utility) rather than focusing on becoming smaller. The underlying message in all of the SFG curriculum is that health is not determined by one’s body size and/or shape. The program teaches adolescent girls how to love their bodies by embracing their inner and outer strength [17,18]. Additionally, the girls participated in activities focused on anatomy, nutrition (focusing on teaching them about balance, variety, and moderation), bullying, establishing a healthy relationship with social media, and ways to promote self-love and positive body image.

SDT served as the foundational theory that guided the program activities and the management of the contextual elements of the program [38]. In addition to creating lessons to improve outcome variables (body image, self-esteem, and physical activity enjoyment), the SFG creators were intentional about enhancing the three primary components of SDT: Competence, Autonomy, and Relatedness. Recent evidence demonstrates that girls who participate in SFG are more self-determined and experienced enhanced competence, autonomy, and relatedness [39].

The first component of SDT, competence, depends on developing and demonstrating a mastery of skills. In addition to being physical active during the workouts at each SFG session, the participants were taught introductory level fitness programming (e.g., circuits versus supersets), modifications to make exercises more challenging or easier, and proper exercise form to ensure safety. At the end of the program, the girls host a family workout night, where they are responsible for creating and leading their family members though a workout. SFG was designed intentionally to build competence by exposing girls to various skill sets and knowledge, providing opportunities for individualized practice with adult feedback, followed by participation in a social environment with their families.

Autonomy, a second pillar of SDT, is defined as a state of personal independence and self-determination. Many of the activities within the SFG program are autonomously driven. For example, one of the SFG activities provides participants with the opportunity to create their own exercise-based game (which they will lead/teach to one another), thereby applying the knowledge gained during the SFG program in a fun, engaging manner. This is a good example of how the program can be designed to support activities that give participants the right balance of what they need (i.e., autonomy and guidance). The third pillar of SDT, relatedness, is the sense of being valued by others and feeling a sense of belonging, both of which are important to adolescent development. During SFG, the coaches create an environment of “sisterhood”, where girls from many different social groups learn to support one another, despite their perceived differences. This network can provide the framework for participants to develop a sense of relatedness, both with the coaches as well as other SFG participants. The community of SFG participants allows for social support from salient sources (e.g., their peers). Adolescents value the opinions and perceptions of their peers more than any other age group [40,41]. The larger the support network of like-minded peers, the more likely a new identity of being a “smart fit girl” is confirmed.

### 2.3. Body Image

To assess participants’ body image, a modified version of the Body-Esteem Scale for Adolescents and Adults (BESAA) was used [42]. The original BESAA consists of three body esteem (BE) subscales: BE-Appearance, BE-Weight, and BE-Attribution [42]. The BE-Appearance section includes 10 statements about the participants’ general feelings about their appearance (e.g., “I like what I look like in pictures”), the BE-Weight section includes eight statements about the individual’s satisfaction with their weight (e.g., “I’m proud of my body”), and the BE-Attribution section includes five statements about perceived evaluations from others about one’s body appearance (e.g., “I’m as nice looking as most people). Emerging evidence suggest that two items in the BE-Attribution subscale may not be appropriate for adolescents (“My looks will help me get a job” and “My looks will help me get dates”) and another BE-Attribution item (“I’m as nice looking as most people”) fits more clearly with the BE-Appearance subscale [43]. As such, the modified BESAA scale was used for this study [43]. It includes all BE-Appearance and BE-Weight statements, and one statement from BE-Attribution (“I’m as nice looking as most people”). The modified BESSA has shown high internal consistency and reliability among middle school girls (BE-Appearance *a* = 0.90 and BE-Weight *a* = 0.93) [43]. Participants answered each statement using a 5-point Likert scale ranging from never (1) to always (5). Items that were negatively worded were reverse coded to reflect a positive body assessment and mean composite scores for BE-Appearance and BE-Weight were created and analyzed. A recent systematic review of body image measures reported that the BESAA scale is both valid and reliable [44]. In the present study, sample, BE-Appearance (*a* = 0.83 and *a* = 0.85) and BE-Weight (*a* = 0.91 and *a* = 0.92) demonstrated high reliability for both pre- and post-data, respectively.

### 2.4. Self-Esteem

The Rosenberg Self-Esteem Scale, which consists of a 10-item scale, was used to assess participants’ self-esteem [45]. Participants were asked statements regarding their self-esteem (e.g., “On the whole, I am satisfied with myself”, “I feel that I have a number of good qualities”). The scale uses a 4-point Likert scale ranging from strongly disagree (1) to strongly agree (4). Items that were negatively worded were reverse coded to reflect positive self-esteem. It has been validated among most age groups, including middle-school-aged children [46]. This measure demonstrated high reliability in the present analysis for both pre- and post-data, respectively (*a* = 0.90 and *a* = 0.91). Mean scores were computed and used for statistical analysis.

### 2.5. Physical Activity Enjoyment

The revised Physical Activity Enjoyment Scale (PACES) was used to assess the participants’ physical activity enjoyment, which has been validated and modified for use among adolescent girls [47]. The revised version consists of 16 rather than 18 statements that begin with “When I am active…” [47]. A 5-point Likert scale, rather than the original 7-point scale, was used with responses ranging from disagree a lot (1) to agrees a lot (5). Items that were negatively worded were reverse coded to reflect a positive physical activity enjoyment. A total score was then calculated by taking the sum of the 16 items. The current analysis demonstrated high reliability for this measure in both the pre- and post-data collection, respectively (*a* = 0.96 and *a* = 0.96).

### 2.6. Statistical Analyses

Data were analyzed using IBM SPSS Version 25. Multiple imputation was used to account for missing data, allowing for unbiased estimate results and a more complete data set. The majority of study variables had less than one percent missing data, with the exception of the Rosenberg Self-Esteem Scale (23.6%). The greater amount of missing data for the Rosenberg Self-Esteem Scale was due to an error that occurred during data collection (i.e., the scale was accidently omitted from the electronic survey) in the spring of 2016 which resulted in that scale not being included as part of the larger questionnaire. Correlation statistics were calculated for all dependent variables. Values between 0.3 and 0.7 indicated a moderate relationship and values between 0.7 and 1.0 indicated a strong relationship [48]. A one-way repeated measures MANOVA is the recommended statistical analysis when determining differences in multiple dependent variables between treatments [49]. Therefore, a repeated measures MANOVA was used to assess cohort differences in body image, self-esteem, and physical activity enjoyment between pre- and post-measurements. Assumptions were reviewed for the repeated measures MANOVA using the Shapiro–Wilk test and tests of skewness and kurtosis for normality, and Levene’s test for equal variances. All assumptions for the statistical test were met; therefore, it was appropriate to statistically analyze the two different sample sizes [50].

## 3. Results

At the initiation of the study, there were no statistically significant differences between the cohorts (*p* > 0.05), indicating no significant variation. Correlation statistics indicated a moderate positive relationship between physical activity enjoyment and BE-appearance, physical activity enjoyment and BE-weight, and physical activity enjoyment and self-esteem. A strong positive relationship existed between BE-Appearance and BE-weight, BE-appearance and self-esteem, and BE-weight and self-esteem. All correlation statistics can be found in Table 1. Descriptive statistics, including mean values, are displayed in Table 2. Percent increases in scores were greater for the intervention group as compared to the control group for all characteristics measured.

A repeated measures MANOVA test was conducted to test intervention effect on body-esteem, self-esteem, and physical activity enjoyment. The results for both the original dataset and for the 5-imputation analysis are reported in Table 3 and Table 4. The multivariate tests demonstrated a main effect of the intervention on the combined dependent variables over time, *F* = 3.24, *p* = 0.013 (Table 3). Univariate tests indicated there was an intervention effect on BE-appearance, *F* = 9.23, *p* = 0.003, and BE-weight *F* = 4.77, *p* = 0.029, but not for self-esteem *F* = 0.26, *p* = 0.611, or for physical activity enjoyment *F* = 2.76, *p* = 0.098 (Table 4). The SFG program resulted in significant improvements from pre to post scores in BE-Appearance and BE-Weight but not for self-esteem or physical activity enjoyment.

## 4. Discussion

This study illustrated how SFG was effective in improving participants’ psychosocial health. While the girls who did not participate in SFG showed slight increases in psychosocial health, adolescent girls who completed the program demonstrated greater, although not all statistically significant, improvements in body image, self-esteem, and physical activity enjoyment. The intervention group saw an increased score from pre to post measures for BE-appearance (11.35%), BE-weight (11.27%), self-esteem (3.22%), and physical activity enjoyment (4.01%) over time. The control group meanwhile saw an increased score over time for BE-appearance (3.22%), BE-weight (2.47%), self-esteem (1.92%), and physical activity enjoyment (0.26%).

Body image was the only construct to exhibit significant differences between the two cohorts. Therefore, the results only partially confirmed our hypothesis. Previous qualitative studies evaluating SFG [17,18] and other research on physical activity programs designed to improve body image and body satisfaction among adolescent girls [51,52] corroborate these findings. The magnitude of the increase in body image was significantly greater among the intervention group for both body image subscales: appearance and weight. This is particularly important to highlight, as SFG participants had lower scores in these categories than the control group before the program began, nearly reaching scores of the control group at post measurements. This suggests that the girls who opted to participate in SFG may have had a greater capacity for improvement than girls in the control group. In this study, appearance was measured based on how the participants felt about their overall appearance. Weight was measured as how satisfied participants were about their weight. The significant improvements seen in these two body image subscales are likely due to SFG’s unique programming that (1) teaches girls about body utility (shifting the importance from body aesthetic to body performance), (2) provides activities and discussions around body image (e.g., lessons on photoshop and how beauty is constructed in society), and (3) creates space to learn about how the media manipulates perceptions of the body [17]. Just as girls can internalize conversations of appearance and the thin ideal [53], this study suggests girls can also internalize conversations on positive body image and body utility.

While self-esteem showed mean increases in both groups, the intervention did not have a significant impact on participants’ self-esteem as it had with body image. Similar results were reported by Smith (2018), who, in a large random control trial of adolescents living in Australia (N = 508), reported no significant impact of a resistance training program on participants’ self-esteem [54]. The magnitude of the increase in self-esteem scores were much smaller than what was seen with body image. Previous literature has found self-esteem to act more stable over a large amount of time among age groups of adolescence and old age [55]. Moreover, recent data suggest that self-esteem demonstrates a developmental pattern characterized by an initial increase from adolescence through middle adulthood, which might be explained by the complex and multidimensional nature of self-esteem [56]. This may be why there was no significant change from pre- to post-measures, as SFG lasts only 10 weeks and there may not have been enough time to see such changes. Further research with longer lasting interventions could be worth investigating for potential changes in self-esteem among adolescent girls. While SFG has a large emphasis on body image and may be why this construct had the greatest increase, the separate measures in this study for body image and self-esteem could have made it more difficult to distinguish the data between the two constructs. Body image is largely related to self-esteem, as these results support, acting as a major contributing factor in an individual’s concept of their self-worth [57]. Improvements in self-esteem could have been overshadowed by the measures of body image.

Additionally, the insignificant change in self-esteem experienced by the intervention group might also be explained by SDT constructs. Most of the activities geared towards providing participants with autonomy, relatedness, and competence were about body image and it is possible that even those activities aimed at improving self-esteem were interpreted as ways to improve body image. For example, one of the activities included a discussion about self-esteem and time for the participants to create positive affirmations. Likely due to society’s hyperfocus on the objectification of girls and women’s appearance [58], the participants would often include affirmations about their body (e.g., you are beautiful) rather than their global self-concept (e.g., you are a kind person).

Similar to self-esteem, there were no intervention effects found over time in physical activity enjoyment scores for girls who participated in SFG. Overall increases in these scores were minimal. Resistance training was the primary modality of physical activity girls participated in during SFG. Despite results from this study, previous research has demonstrated an association between resistance training and physical activity enjoyment. Michael et al. [59] reported that among adolescent girls, team sports/weight-lifting activities were positively associated with physical activity and physical activity enjoyment. Pate et al. [60] reported similar findings with weightlifting and other physical activities that girls enjoyed increased their time spent in vigorous physical activity. However, it is unknown why PACE scores increased for the control group, as Haas, Yang, and Dunton [34] found that enjoyment of physical activity decreased in youth over time, along with moderate-to-vigorous physical activity. It is possible that those in the control group participated in other forms of activities that they found enjoyable, possibly from their physical education classes, thereby increasing their own enjoyment perspectives of physical activity. For the intervention group, the non-statistical increase in physical activity enjoyment may also be a result of only being exposed to one primary form of physical activity (i.e., resistance training) rather than experiencing multiple modalities of physical activity. The lack of statistical differences in physical activity enjoyment could also be a result of the intervention including multiple components outside of physical activity, resulting in less time being physically active (90 min of psychosocial activities and only 30 min of resistance training). Compared to body image activities (which the girls can practice anywhere), it is possible that the girls were only able to demonstrate competence, experience relatedness, and be autonomous with resistance training during the intervention due to lack of access to such equipment at home.

## 5. Strengths/Limitations

There were multiple limitations to this study as well. As this study was conducted in one middle school, it was not possible to determine whether the results can be generalized to other middle schools where SFG has been used. Future research on a broader scale will be needed to test the effectiveness of SFG on physical activity enjoyment, body image, and self-esteem across other settings. Another limitation was the lack of randomization and control of outside activities in the study design. Since SFG is a voluntary after-school program, randomization of groups could not be performed and controlling for the participants leisure time physical activity and psychosocial health activity participation was not possible. An additional limitation was the sparce participant demographic information available to the researchers. While all the data were collected in a similar demographic area, the limited information available to the researchers provided issues when interpreting and generalizing the results. Future research of such programming should also (1) focus on control of all physical activity participated in during program implementation, (2) include a longitudinal design, and (3) include multiple data-collection points throughout program implementation.

This study had several strengths as it sought to identify the efficacy of SFG on adolescent girls’ physical activity enjoyment, body image, and self-esteem. First, data were collected at pre- and post-intervention, allowing researchers to measure the change in variables over time among the intervention and control group. Another strength was the control group with which to compare participants in the intervention. Additionally, previously validated measures among this group were used to gather quantitative data. There was also nearly a one-hundred-percent retention rate among study participants. Additionally, the mean age of both intervention and control groups was almost identical and each group was comprised of girls from the same middle school.

## 6. Conclusions

Participation in SFG resulted in increased body image, specifically how the girls felt about their appearance and weight. This study hopes to encourage future programming tailored to the specific health promotion needs and experiences of adolescent girls (e.g., physical activity, body esteem). The results provide support that intentionally designed programs can be used to increase psychosocial outcomes among adolescent girls. Programs such as SFG can support engagement in physical activity, as body-image, self-esteem, and physical activity enjoyment are important factors for this group. As girls continue to grow and develop, the deliberate design of programs aimed to promote their physical and psychosocial health will be critical.

## Figures and Tables

**Figure 1 behavsci-13-00783-f001:**
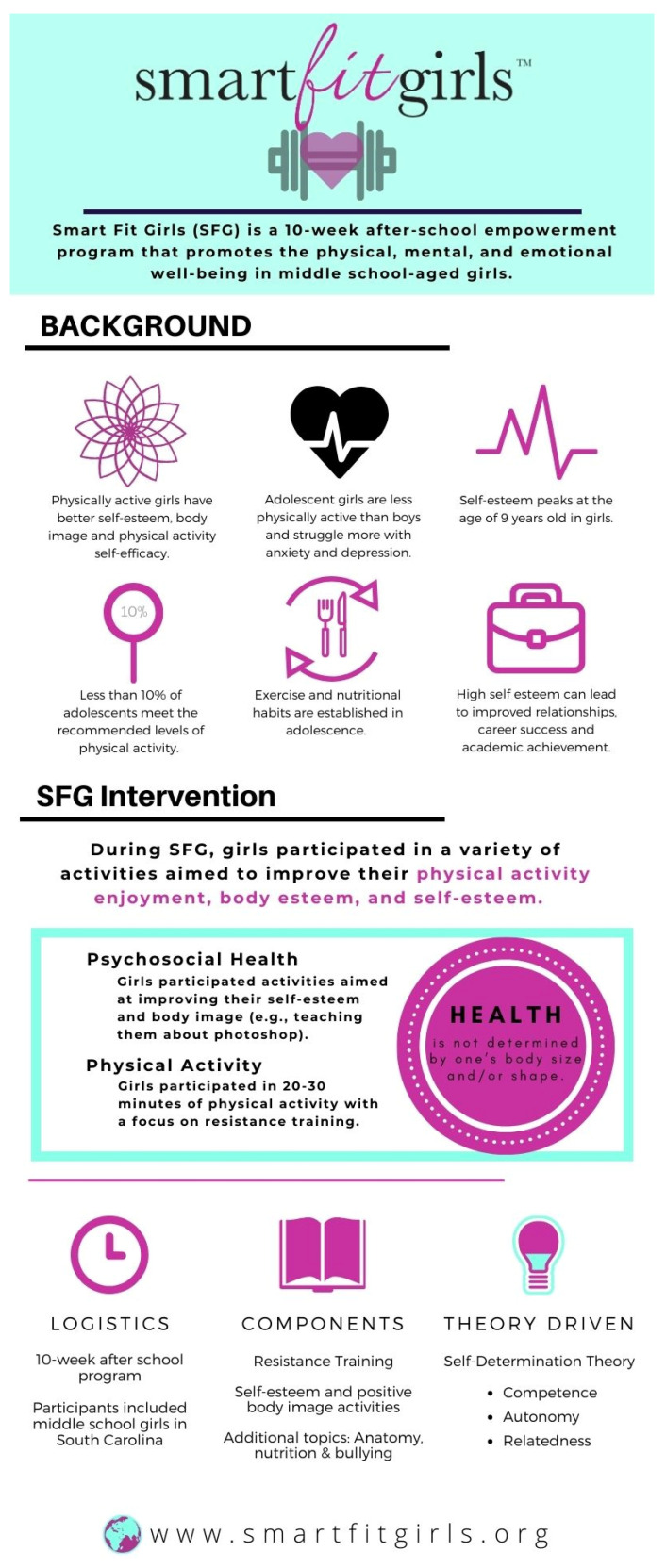
An infographic providing a visual representation of the intervention.

**Table 1 behavsci-13-00783-t001:** Variable correlations.

	Body-Esteem (Appearance)	Body-Esteem (Weight)	Rosenberg Self-Esteem	Physical Activity Enjoyment
Body-Esteem (Appearance)	1	-	-	-
Body-Esteem (Weight)	0.774	1	-	-
Rosenberg Self-Esteem	0.786	0.725	1	-
Physical Activity Enjoyment	0.360	0.304	0.407	1

**Table 2 behavsci-13-00783-t002:** Pre- and post-descriptive statistics for the control and intervention group.

	Control						Intervention				
	Time	N	Min	Max	Mean (+SD)	% Change	N	Min	Max	Mean (+SD)	% Change
Body-Esteem (Appearance)	Pre	401	1.45	4.64	3.11 (0.76)		47	1.36	4.64	2.82 (0.81)	
	Post	401	1.36	4.64	3.21 (0.80)	3.22%	47	1.36	4.64	3.14 (0.83)	11.35%
Body-Esteem (Weight)	Pre	401	1.00	5.00	3.24 (1.04)		47	1.00	5.00	2.75 (1.11)	
	Post	401	1.00	5.00	3.32 (1.03)	2.47%	47	1.00	5.00	3.06 (1.07)	11.27%
Rosenberg Self-Esteem	Pre	401	10.0	40.0	29.13 (6.49)		47	10.0	40.0	27.60 (6.78)	
	Post	401	3.0	40.0	29.69 (6.66)	1.92%	47	10.0	40.0	28.49 (7.43)	3.22%
Physical Activity Enjoyment	Pre	401	1.00	5.00	3.79 (0.89)		47	2.69	5.00	3.99 (0.70)	
	Post	401	1.00	5.00	3.80 (0.90)	0.26%	47	2.19	5.00	4.15 (0.69)	4.01%

**Table 3 behavsci-13-00783-t003:** Multivariate tests for the main effect of the intervention.

Effect	N	*F* (5 Imputation Range)	*p* (5 Imputation Range)	*η* ^2^
Intercept	448 (590)	1255.93 (1643.60, 1651.58)	<0.001 (<0.001)	0.919 (0.918, 0.919)
Intervention Group	448 (590)	4.13 (4.91, 5.05)	0.003 (<0.001)	0.036 (0.032, 0.033)
Time	448 (590)	8.78 (7.07, 7.40)	<0.001 (<0.001)	0.073 (0.046, 0.048)
Time × Intervention Group	448 (590)	3.24 (3.51, 4.67)	0.013 (0.001, 0.008)	0.028 (0.023, 0.031)

**Table 4 behavsci-13-00783-t004:** Univariate tests demonstrating intervention effect for BE-Appearance and BE-Weight.

Effect	Variable	*F* (5 Imputation Range)	*p* (5 Imputation Range)	*η* ^2^
Time	Body-Esteem (Appearance)	32.59 (24.75, 25.58)	<0.001 (<0.001)	0.068 (0.040, 0.042)
	Body-Esteem (Weight)	14.25 (19.99, 20.23)	<0.001 (<0.001)	0.031 (0.033, 0.033)
	Rosenberg Self-Esteem	5.14 (3.02, 9.85)	0.024 (0.002, 0.083)	0.011 (0.005, 0.016)
	Physical Activity Enjoyment	3.94 (1.06, 1.19)	0.048 (0.270, 0.305)	0.009 (0.002, 0.002)
Time × Intervention Group	Body-Esteem (Appearance)	9.23 (9.71, 10.24)	0.003 (0.001, 0.002)	0.020 (0.016, 0.017)
	Body-Esteem (Weight)	4.77 (12.53, 12.68)	0.029 (<0.001)	0.011 (0.021, 0.021)
	Rosenberg Self-Esteem	0.26 (0.00, 4.12)	0.611 (0.043, 0.972)	0.001 (0.000, 0.007)
	Physical Activity Enjoyment	2.76 (0.93, 1.00)	0.098 (0.318, 0.335)	0.006 (0.002, 0.002)

## Data Availability

The data presented in this study are available on request from the corresponding author. The data are not publicly available due to privacy and ethical restrictions.

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
