# Peer review of "The Influence of a Girls’ Health and Well-Being Program on Body Image, Self-Esteem, and Physical Activity Enjoyment"

_behavsci, 2023, doi:10.3390/bs13090783_

Round 1
Reviewer 1 Report
I thought the paper was well done as is and would add to the scholarship on this topic. Strategies to address body image and self-esteem are important and needed. Introduction of ideas, like the authors did here, are likely to be of interest to readers and practitioners alike. I thought this paper was well written, clear, and concise. Additionally, it was grounded in previously conducted scholarship. That said, I really appreciated the practical implications. This issue is a challenge faced by many. Sharing ideas of strategies will help scholars and practitioners continue to make improvements in the field which will positively impact those struggling with body image and self-esteem. I don’t have suggestions for improvement. I truly thought it read very well.
Author Response
Thank you for the thoughtful and kind feedback. We appreciate the time you took to review our paper.
Reviewer 2 Report
Dear Authors.
We appreciate the effort made in developing this work and the contribution it makes to the field of physical activity. In any case, I have to make some contributions that I believe are very relevant for the improvement of the work and that can be seen below.
Abstract: I think it would be interesting to give some notion about the characteristics of the intervention. Otherwise, it is difficult to interpret what is intended to be referred to at the end, when talking about psychosocial lessons. More concretely, the relevance given to SDT and motivation can be appreciated throughout the work, so they could be relevant points to consider as incorporations.
Introduction (22-27): The way in which the benefits associated with the practice of physical activity are presented is strange. Why do you only talk about the physical and cognitive area and then specify the risk of depression? I understand that it has to do with the fact that it is something that is referred to later, but, in that case, I believe that reference to all the realities alluded to should be included.
Self-esteem (83): The definition of such an important aspect, in the article, as self-esteem, should be justified. Which authors are those who take it into consideration as it is expressed in the work; (85-87): I do not quite understand the reference to physical changes when speaking of a much broader concept that occupies higher hierarchical positions, such as self-esteem.
1.3. Physical activity enjoyment (95): should not be the last line on a page. I think I understand the intended relationship between enjoyment and intrinsic motivation, but the latter appears "out of the blue", and it is necessary to establish a relationship that can clarify everything for the reader.
Self-Determination Theory: perhaps point 1.4. should be presented before 1.3., so that the link between enjoyment in the game and motivation can be appreciated more clearly and, of course, establishing a relationship in the passage from one section to the other; if relevance is given not only to the practice of physical activity, but also to its enjoyment, perhaps this enjoyment should also be considered here.
2.1. Sample: aspects related to the evaluation instruments should be omitted, since they already have a specific section for their treatment; how is it possible that out of a sample of 590 girls, only 58 are incorporated into the intervention? I believe that a great opportunity to incorporate a much larger sample into the intervention is lost. I think that with these data it would be possible to justify, in some way (SPSS, G*Power...) the size that the authors indicate. In addition, I think it is necessary to detail the reasons why the difference is so large, since only inclusion criteria are mentioned, but not what they are. I think this is relevant.
2.2. Intervention: the quotation of lines 161-162 is not correct; should it be understood that they are replicating a study that has already been done? In that case it would be necessary that, at some point (objectives to be achieved, discussion, conclusions) it is mentioned what the present work contributes with respect to the original; The physical condition work seems to be the least accepted by the students, which could cause contradiction in relation to the enjoyment. How can this fact be taken care of? Are there other types of proposals? If so, I think it is important to mention them in order to clarify the program; At this point (172-176), I find it relevant that, in the introduction, the benefits of the practice of physical activity are not mentioned exclusively, since the intervention incorporates other health-related aspects that, in fact, take up more time than physical activity; The definitions related to the constituent elements of TDS should be incorporated in the corresponding section, so that, here, the reader could focus exclusively on the characteristics of the intervention program.
2.3. Body image: although this is less important, perhaps it would be more coherent to mention a heading related to variables and instruments used for their evaluation and, within it, to mention them; Do the modifications of the instrument (BESAA) have to do with the inadequacy of any of the items? Why? Is an instrument whose psychometric properties are not adequate for the study population being used? Now mention is made of pre-adolescents, but previously, the authors always referred to adolescents. Something is not right. And, after all this, hasn't an instrument been found that matches the needs associated with the characteristics of the sample?
2.5. Physical activity enjoyment: I think it is better to mention only the instrument used and not other previous versions.
Statistical analyses: I do not understand the reasons for incorporating scales once the research has started.
Are different amounts of data incorporated for each instrument?
3. Results: What are the initial data for the groups; are there significant differences; statistics should be incorporated after referring to correlations; why do they refer to score increases in percentages and do not mention if there are significant improvements or not? Likewise, it would be interesting to see if there are significant differences between groups (intervention and control).
4. Discussion: Why were only differences in body image found between the two cohorts? I do not think that qualitative studies can be compared with a quasi-experiment like this one; what about the rest of the variables; is the ceiling effect taken into account in this case? The baseline data of both groups could give some relevant information in this regard; how do we know if it is due to the practice of physical activity, to the information alluded to or to the effect derived from the combination of both aspects? There are studies that give better results to interventions based solely on physical activity (https://www.tandfonline.com/doi/abs/10.1080/02701367.2021.1927945?tab=permissions&scroll=top); Quizá, el hecho de que la autoestima ocupe una posición jerárquica superior, haya hecho que sea más estable, además de que depende de muchos más determinantes que la imagen corporal; Parece poco coherente que uno de los aspectos relevante de la intervención sea el disfrute de la actividad física y se incorporen actividades que no se asocian a ese disfrute. ¿Qué sentido tiene? ¿Por qué no se da esa variedad de actividades de la que se habla?
Author Response
Please see our responses in red in the attached document.

Reviewer 3 Report
General comment: The present study has an interesting theme and verifying possible effects of body image programs for self-esteem is important. However, some aspects that can be improved in the present study are reported below.
Comment: The authors did not mention the innovative aspects of the present study in the introduction. When compared to previous studies, what is new in this study?
Comment: What is the hypothesis of this study?
Comment: The introduction should not be fragmented as it is in the present study, but presented in a single section. I suggest the authors put all the elements in a single part.
Comment: Enter information about the sample calculation.
Comment: Describe more clearly how the participants were recruited.
Comment: Regarding the intervention, I suggest authors create an infographic. This would make the information more didactic and easier to understand.
“This study illustrated how SFG was effective in improving participants’ psychosocial health. While the girls who did not participate in SFG showed slight increases in psycho-social health, adolescent girls who completed the program demonstrated greater, although not all statistically significant, improvements in body image, self-esteem, and phys-301 ical activity enjoyment. Body image was the only construct to exhibit significant differences between the two cohorts. Previous qualitative studies evaluating SFG [11, 12] and other research on physical activity programs designed to improve body image and body satisfaction among adolescent girls [49, 50] corroborate these findings. The magnitude of the increase in body image was significantly greater among the intervention group for both body image subscales: appearance and weight. This is particularly important to high light as SFG participants had lower scores in these categories than the control group before the program began, nearly reaching scores of the control group at post measurements. This suggests that the girls who opted to participate in SFG may have had a greater capacity for improvement than girls in the control group. In this study, appearance was measured based on how the participants felt about their overall appearance. Weight was measured as how satisfied participants were about their weight. The significant improvements seen in these two body image subscales are likely due to SFG’s unique program ming that teaches girls about 1) body utility (shifting the importance from body aesthetic 315 to body performance), 2) provides activities and discussions around body image (e.g., les-316 sons on photoshop and how beauty is constructed in society), and creates space to learn about how the media manipulates perceptions of the body [11]. Just as girls can internalize conversations of appearance and the thin ideal [51], this study suggests girls can also internalize conversations on positive body image and body utility”.
Comment: Comment: The authors bring interesting information in the paragraph mentioned above. However, it remains to discuss the reasons and possible mechanisms for the findings observed in the present study.
While self-esteem showed mean increases in both groups, the intervention did not have a significant impact on participants’ self-esteem as it had with body image. Similar results were reported by Smith (2018) ,who in a large random control trial of adolescents living in Australia (N = 508), reported no significant impact of a resistance training program on participants’ self-esteem[52]. The magnitude of the increase in self-esteem scores were much smaller than what was seen with body image. Previous literature has found self-esteem to act more stable over a large amount of time among age groups of adoles- cence and old age [53]. This may be why there was no significant change from pre- to post- measures as SFG lasts only 10 weeks and there may not have been enough time to see such changes. Further research with longer lasting interventions could be worth investi gating for potential changes in self-esteem among adolescent girls.
Comment: Comment: Could physical activity performed during leisure time by the control group also influence these findings? Perhaps discussing this point would also help clarify this paragraph.
Comment: The limitations are pointed out very well by the authors. However, I suggest including the strengths after the limitations.
Comment: What are the practical applications of this study?
Comment: What are the suggestions for future studies?
Author Response
See responses in red in the attached document.

Round 2
Reviewer 2 Report
Dear authors, I believe that an important effort can be appreciated in trying to improve the proposed work, although there are some aspects that have not been dealt with in depth or have not been considered.
Abstract: a very small reference is added to SDT as a determinant of the intervention, without being specific enough to know which aspects are relevant and what they contribute to the intervention. I believe that not much has changed from what was established at the beginning.
Introduction: references are eliminated, when what is requested is exactly the opposite. I believe that what should be done is to complete and justify, not to eliminate.
Section 1.3: remains as the last line of the page, although it now corresponds to another point. It has not been modified. On the other hand, nothing of what is indicated is answered.
Section 2.1. Sample: I still do not understand the reasons for establishing such a disparate design, regardless of what the authors argue. Perhaps it would be necessary to justify what leads to this.
Section 2.2. Intervention: the authors say that they do not know what leads me to think that a study has been replicated. The fact that they talk about the intervention as a structured program that corresponds to Walters, Chard, Jordan and Anderson does. Something that now, moreover, can be seen in Figure 1; nothing is answered in relation to the mention of the consideration that students can make of physical fitness work, confusing physical fitness with physical activity; I must insist that, in addition to what is related to the practice of physical activity and SDT, mention is made of a good number of aspects that are not considered in any way, something that is missing (anatomy, nutrition, bullying, establishing a healthy relationship with social media, and ways to promote self-love and positive body image).
Section 2.3. Body image: what is obviated is what is related to the way of referring to the participants throughout the work (adolescents - preadolescents).
Statistical analysis section: the authors are confused by the reference to the incorporation of scales. I don't know if I misunderstand, but they are the ones who indicate that the missing data related to the Rosenberg Self-Esteem Scale is because that part of the questionnaire was not included in the spring of 2016.
Results section: there is no answer to whether there are significant differences between groups at the beginning; the tables should be named referring to the "relationships" between variables and not to the test used, for clarity; regardless of the relevance that a small change may have, in the results section, after a statistical analysis, what is expected is to appreciate the data found after such analysis, without, sincerely, percentages being able to have a place. Apart from this, improvements, even if they are not statistical, can be referred to in the discussion, conclusions, etc. section;
Discussion section: the qualitative studies that are used to compare with this quasi-experiment are different studies... On the other hand, it is true that it is complex to know what "weight" each part of the intervention has in the results found, but perhaps it could be understood as one of the most important aspects to consider and, finally, the part that I copied in Spanish language, said the following: "Perhaps, the fact that self-esteem occupies a higher hierarchical position, has made it more stable, in addition to the fact that it depends on many more determinants than body image; It seems inconsistent that one of the relevant aspects of the intervention is the enjoyment of physical activity and activities that are not associated with that enjoyment are incorporated. What is the point, and why isn't there that variety of activities being talked about?".
Regards.
